# Integrated Transcriptome and Biochemical Analysis Provides New Insights into the Leaf Color Change in *Acer fabri*

Guohua Liu [1,†], Heng Gu [2,†], Hongyu Cai [1], Congcong Guo [1], Ying Chen [3], Lianggui Wang [2,*] and Gongwei Chen [1,*]

1   Jiangsu Vocational College of Agriculture and Forestry, Jurong 212400, China; liuguohua@jsafc.edu.cn (G.L.); m13912109566@163.com (H.C.); guocongcong0110@163.com (C.G.)
2   College of Forestry, Nanjing Forestry University, Nanjing 210037, China; guheng@njfu.edu.cn
3   Liyang Comprehensive Agricultural Technology Extension Center, Liyang 213311, China; baiye_118@163.com
*   Correspondence: wlg@njfu.com.cn (L.W.); chengongwei0118@163.com (G.C.)
†   These authors contributed equally to this work.

**Abstract:** *Acer fabri* is a widely distributed ornamental tree with colorful leaves and high ornamental value. Its young leaves change from red to red and green until turning fully green. To understand the mechanism of leaf color change, transcriptome sequencing and pigment content determination were performed in three stages during the leaf color change of *A. fabri*. In total, 53,550 genes, including 838 transcription factors (TFs), were identified by transcriptome sequencing. In addition, the results of orthogonal partial least squares-discriminant analysis (OPLS-DA) of three pigments in the three stages of leaf color development suggested that carotenoids played a major role in the process of leaf color change from red to red-green, whereas anthocyanins played an important role in the process of leaf color change from red to green. Based on weighted gene co-expression network analysis (WGCNA), *Af0034384* (HSFB2A), *Af0051627* (NMT1), and *Af0052541* (THY-1) were selected as hub genes from characteristic modules with significant correlation between carotenoids and anthocyanins. The results of gene network regulation maps and real-time fluorescence quantitative PCR (qRT-PCR) showed that *Af0010511* (NAC100) upregulated the expression of *Af0034384* (HSFB2A), leading to an increase in carotenoid content and the gradual greening of leaves during the transition from red to green. However, during the transition from red to green leaves, *Af0033232* (NAC83) and *Af0049421* (WRKY24) downregulated the expression of *Af0051627* (NMT1) and *Af0052541* (THY-1), respectively, leading to a decrease in anthocyanin content and the complete greening of leaves. These results could provide new ideas for studying the molecular mechanism of leaf color change in *A. fabri* and other species.

**Keywords:** *Acer fabri*; leaf; pigment; transcriptome; WGCNA





## 1. Introduction

Plants with leaves that change color over time exist widely in nature and have unique ornamental value [1,2]. According to the appearance of leaf color in different periods, they can be divided into spring, autumn, and normal leaf plants. However, the mechanism of leaf color change is complex and has thus become a research focus. Studies on a variety of colorful-leaved plants have found that the change in leaf color is caused by the type and content of pigments in their tissues [3]. Chlorophyll, carotenoids, and anthocyanins are the main pigments that cause leaf color changes. When the proportion of these pigments in leaves changes significantly, the color of the leaves also changes [4,5]. Chlorophyll is produced in chloroplasts and is an important photosynthetic pigment [6]. The green color of plant leaves mainly results from a high chlorophyll content, which is determined by the balance between chlorophyll synthesis and degradation. Carotenoids are an important class of fat-soluble pigments which exist widely in plants. As secondary metabolites, carotenoids

render flowers, leaves, and fruits brightly colored [7]. Anthocyanin is a common water-soluble pigment in plants and has a range of functions. It is found in leaves, flowers, and fruits, rendering them a variety of colors, including red, purple, and blue [8–10]. Therefore, it is necessary to identify the key pigments and regulatory pathways involved in leaf color changes to determine the resulting colors.

In recent years, the rapid development of sequencing technology has reduced the time and cost of sequencing and has been widely used in transcriptome research. Transcriptome sequencing can be used to quickly and effectively differentially analyze expressed genes (DEGs) and to provide a functional annotation of regulatory pathways [11]. This method has been widely used in the study of leaf color change. For example, transcriptional sequencing was conducted on different-colored leaves of *Acer rubrum* to analyze the molecular mechanisms involved in anthocyanin biosynthesis and accumulation, thus revealing the mechanisms involved in the formation of red *A. rubrum* leaves [3]. The transcriptome sequencing of green leaves of *Osmanthus* identified key genes involved in regulation at different developmental stages; this, combined with the analysis of related physiological indicators, suggested the mechanisms involved in osmanthus leaf color determination [12]. In addition, in a study of *Ginkgo biloba*, DEGs and transcription factors (TFs) involved in chloroplast development and pigment biosynthesis were identified by the transcriptome sequencing of golden leaves and normal green leaves, providing insights into the molecular mechanisms of yellowed plant phenotypes [13]. However, there is no report of the molecular mechanisms involved in leaf color change in the popular ornamental tree *Acer fabri*.

Leaf color change is a complex process regulated by many genes. The transcriptome sequencing of three different species of chrysanthemum detected TF *CmNAC73* and showed it to be involved in chlorophyll biosynthesis and the regulation of leaf color change in chrysanthemum [14]. In a study of *Populus laurifolia*, the expression level of TF *PdbHLH143* was higher in the leaves of *P. laurifolia* than in other tissues, suggesting that TF *PdbHLH143* is involved in the biosynthesis of anthocyanin in the leaves [15]. Similarly, in studies of the color of green leaves of *Osmanthus fragrans* 'Yinbi Shuanghui' in different developmental stages, key genes (TF *WRKY3* and TF *WRKY4*) involved in leaf color change were revealed [12]. In addition, a recent study showed that TF *AcMYB2* is a TF from *Aglaonema commutatum* 'Red Valentine' that regulates anthocyanin biosynthesis, and its overexpression leads to a red leaf phenotype in transgenic plants [1]. However, the genes related to leaf color change in *A. fabri* are unknown, and the molecular mechanisms involved remain unclear.

*A. fabri* is a colorful ornamental tree widely distributed in southern China [16]. It is a beautiful tree with luxuriant foliage, especially its leaves, which transition from red to red and green until turning completely green. Given its abundant leaf color variation, especially in autumn, the tree is increasingly used for landscaping in subtropical areas [17]. At present, studies on *A. fabri* mainly focus on its introduction, domestication, and rapid propagation, with no studies of the leaf color change mechanisms [18]. Therefore, in this study, pigment content analysis and transcriptome sequencing were conducted in the three stages of leaf color development in *A. fabri*. The key genes that regulate the color change of *A. fabri* leaves were screened by a coexpression network analysis (WGCNA) of pigment content and transcriptome. This study lays a foundation for elucidating the molecular mechanism of leaf color change in *A. fabri*.

## 2. Materials and Methods

### 2.1. Plant Materials

The plant material used in this experiment was *A. fabri*. The collection of plant material complied with relevant institutional, national, and international standards and regulations. The *A. fabri* used in the study was a 5-year-old plant of a colorful leaf nursery stock base, growing in Jiangsu Agricultural Expo Garden (32°01′ N, 119°24′ E). The leaf development of *A. fabri* can be divided into three stages (Figure 1), namely the young leaf (red) and

the transitional (red and green) and mature stages (green). Leaves with the same growth pattern were all sampled at 10:00 h on a sunny day. The collected leaves were placed into sterile tubes, immediately frozen in liquid nitrogen for 10 min, and then transferred to a refrigerator at −80 °C for preservation. Three biological replicates of leaves at each developmental stage were performed.

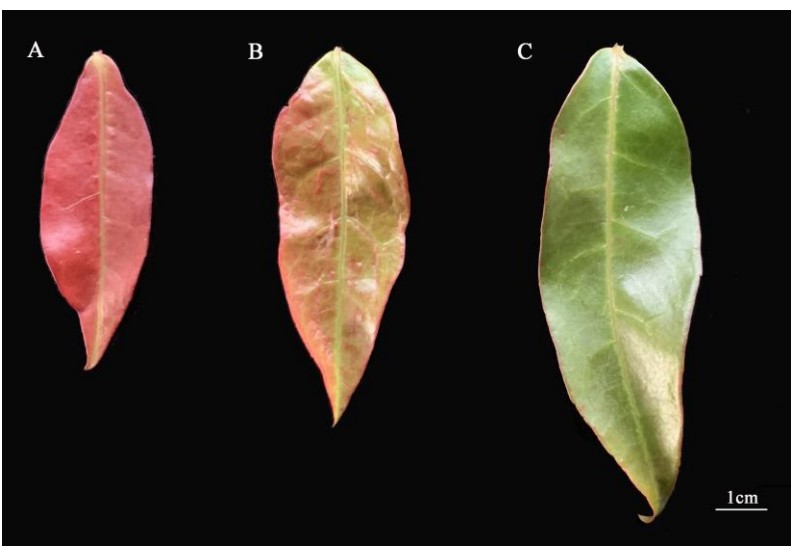

**Figure 1.** Different developmental stages of an *A. fabri* leaf: (**A**) red leaf; (**B**) red and green leaf; and (**C**) green leaf.

## 2.2. Determination of Pigment Content

Pigment content was determined using the methods described by Zhang and Chen [12,19]. The ground leaves were soaked with 95% ethanol for 24 h in the dark. Chlorophyll and carotenoid contents were determined in a spectrophotometer at 470, 649, and 665 nm. Anthocyanins were determined by using a plant anthocyanin kit (Jiancheng, Nanjing, China).

## 2.3. RNA Extraction, cDNA Library Preparation, and Sequencing

Total RNA was extracted from the nine samples using an RNA purification kit (Invitrogen, Carlsbad, CA, USA). The integrity of the RNA was verified by RNase-free agarose gel electrophoresis, and its concentration was measured using a Nano Drop 2000 spectrophotometer (Thermo Fisher Scientific, Waltham, MA, USA). The enriched mRNA was then fragmented into short fragments using fragmentation buffer and reverse transcribed into cDNA with random primers. Second-strand cDNAs were synthesized by DNA polymerase I, RNase H, dNTP, and buffer. The cDNA fragments were then purified with a QiaQuick PCR extraction kit (Qiagen, Venlo, the Netherlands) and end-repaired; poly(A) was then added, and the fragments were ligated to Illumina sequencing adapters. The ligation reaction was purified with AMPure XP Beads (1.0×). The ligated fragments were subjected to size selection by agarose gel electrophoresis and amplified by polymerase chain reaction (PCR). For each stage of *A. fabri* leaf color change, three RNA samples were used to construct a cDNA library and for Illumina Novaseq6000 sequencing, which was completed by Gene Denovo Biotechnology Co. (Guangzhou, China). The transcriptome data were uploaded to the NCBI Sequence Read Archive (https://www.ncbi.nlm.nih.gov/sra/ (accessed on 27 February 2023)) under accession number PRJNA 939819.

## 2.4. De Novo Assembly of RNA-Seq Reads and Quantification of Gene Expression

To obtain more accurate gene expression results, we removed the adapter sequence from the original read and used fastp 0.18.0 version (https://github.com/OpenGene/fastp (accessed on 7 December 2022)) from each data set to delete low-quality reads (mass fraction in the following 10 bases of >40% and/or unknown bases of >10%). Clean, high-

quality sequencing reads from all samples were then combined. Using HISAT 2.2.4 (https://daehwankimlab.github.io/hisat2/ (accessed on 7 December 2022)), the ends of pairs of clean reads were mapped to a reference genome [20]. The reads were mapped through the method based on the reference of each sample to the String Tie version 1.3.1 assembly (http://ccb.jhu.edu/software/stringtie/ (accessed on 7 December 2022)) [21,22]. Based on the number of unique mapping reads, the gene expression level was measured by the exon model fragments per kilobase of transcript per million mapped reads (FPKM) to eliminate the influence of different gene lengths and sequencing differences on the calculation of gene expression [11]. Genes with a fold change $\geq 2$ or $\leq -2$ and a Q value $\leq 0.05$ are considered to be significantly differentially expressed during plant development [23].

### 2.5. Screening of Hub Genes

All gene sequences were compared with the Gene Ontology (GO) and Kyoto Encyclopedia of Genes and Genomes (KEGG) databases [24–26]. The threshold was set to E-value $< 10^{-5}$ to obtain gene function information to further reveal the biological function of these genes [27–29]. Subsequently, the content of each of the three pigments was compared with DEGs for WGCNA. Genes with similar expression patterns were put into the same module, and cluster trees were constructed according to the gene expression level and correlation of modules. We identified the feature modules of interest and determined the feature module values (MM > 0.9) to further identify hub genes. In order to understand the relationship between hub genes and marginal genes, we sorted downstream genes from high to low according to the weight values. According to the weight values (>0.2), TFs were screened-out, and the Cytoscape software was used to map the regulatory network.

### 2.6. Verification of Gene Expression Using qRT-PCR

The selected key genes and TFs were verified by real-time quantitative (qRT)-PCR using an SYBR Premix Ex Taq kit (Takara Biotechnology, Nanjing, China) as follows: reaction at 95 °C for 3 min, then at 95 °C for 5 s, followed by 60 °C for 30 s, with 40 cycles. The Primer 5 software was used to design the primers (Table S1). TUB from *Acer buergerianum* was selected as the internal reference gene [30]. RNA extracted from maple leaves was reverse-transcribed into cDNA by SuperMix (Transgen, Beijing, China) and diluted 20 times for gene expression tests [31]. Relative gene expression was calculated by using the $2^{-\Delta\Delta Ct}$ comparative Ct method [32]. Three technical replicates and three biological replicates were used for each sample.

## 3. Results

### 3.1. Leaf Pigment content of A. fabri

The contents of three pigments (chlorophyll, carotenoids, and anthocyanins) were determined in the leaves of *A. fabri* at different developmental stages (Figure 2A). During the transition from red to red and green to green, the chlorophyll and carotenoid contents both showed an increasing trend, whereas the anthocyanin content decreased as the leaves gradually turned green. In addition, we used the Simca14.1 software for orthogonal projections to latent structures discriminant analysis (OPLS-DA) of the three pigments in the *A. fabri* leaves. The data points of the three pigments all fell within the 95% confidence interval (CI). The clustering of biological repetitive data points in the same period was clear, and the clustering of relevant data points in different regions at different developmental stages was obvious (Figure 2B). According to the maximum value of the projection variable (VIP), carotenoids were the main pigments in leaves turning from red to red-green, whereas anthocyanins were the main pigments in leaves turning from red to green (Table S2).

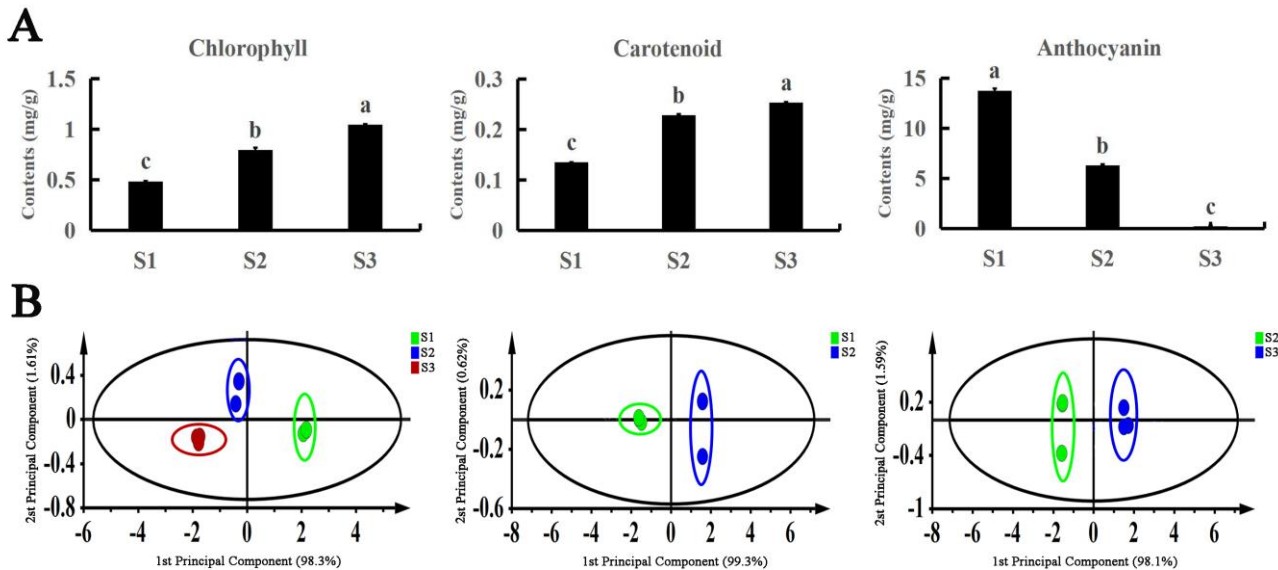

**Figure 2.** Analysis of pigment content in *A. fabri* at different developmental stages. (**A**) Content of the three pigments content in *A. fabri* at different developmental stages. Different letters denote significant differences according to the Tukey's test ($p < 0.05$). (**B**) The score plot of pigment content in *A. fabri* at different developmental stages using OPLS-DA analysis.

### 3.2. Transcriptome Sequencing Results

Nine samples from three developmental stages of maple leaves were sequenced, and a total of 428,385,062 raw sequencing reads and 425,892,206 clean sequencing reads were obtained (Table 1). The GC content of each sample was >45%, the Q20 content was >95%, and the Q30 content was >88% (Table 1). In total, 53,550 genes were identified, including 838 TFs from 42 TF families (Figure S1).

**Table 1.** Summary of sequencing data.

| Sample | Raw Reads | Clean Reads | GC (%) | Q20 (%) | Q30 (%) |
|---|---|---|---|---|---|
| S1-1 | 45,413,200 | 45,050,476 | 44.50 | 95.18 | 88.47 |
| S1-2 | 50,659,672 | 50,336,968 | 44.48 | 96.25 | 90.39 |
| S1-3 | 54,499,748 | 54,159,446 | 44.90 | 95.89 | 89.74 |
| S2-1 | 44,488,688 | 44,168,366 | 44.61 | 95.93 | 89.82 |
| S2-2 | 47,964,364 | 47,662,154 | 44.35 | 96.23 | 90.36 |
| S2-3 | 55,905,296 | 55,525,380 | 45.14 | 95.84 | 89.66 |
| S3-1 | 41,864,648 | 41,720,592 | 44.78 | 97.67 | 93.34 |
| S3-2 | 43,964,838 | 43,819,328 | 44.39 | 98.12 | 94.32 |
| S3-3 | 43,624,608 | 43,449,496 | 44.48 | 97.49 | 92.93 |

S1: red leaf; S2: red and green leaf; S3: green leaf. Raw reads: original number of reads obtained by sequencing; Clean reads: number of reads after removing low-quality reads and trimming adapter sequences; GC%: percentage of G and C in total bases; Q20: Phred score, indicates 99% accuracy of sequenced bases; Q30: Phred score, indicates 99.9% accuracy of sequenced bases.

### 3.3. Hub Genes Were Screened by WGCNA

RNA-seq was used to analyze DEGs during leaf color change. To further identify the key genes involved in the color change of *A. fabri*, the contents of three pigments and their transcriptomes were analyzed by WGCNA. In total, 17 co-expression modules were obtained by weight values, and each module contained a specific number of genes (Figure 3). First, the upregulated characteristic module '17' corresponding to carotenoid was identified, and three key genes with module membership (MM) values > 0.9 were selected: *Af0005512*, *Af0023447*, and *Af0034384*. Then, qRT-PCR was performed on the selected genes, and two genes consistent with the trend in the transcriptome data were selected as hub genes: *Af0023447* and *Af0034384* (Figure 4). Similarly, five hub genes (*Af0006119*, *Af0006650*,

*Af0022301*, *Af00051627*, and *Af0052541*) were screened out in the upregulation characteristic module '10' corresponding to anthocyanins. Based on the correlation between expression trends of qRT-PCR and transcriptome data, two final hub genes, *Af00051627* and *Af0052541*, were identified (Figure 4).

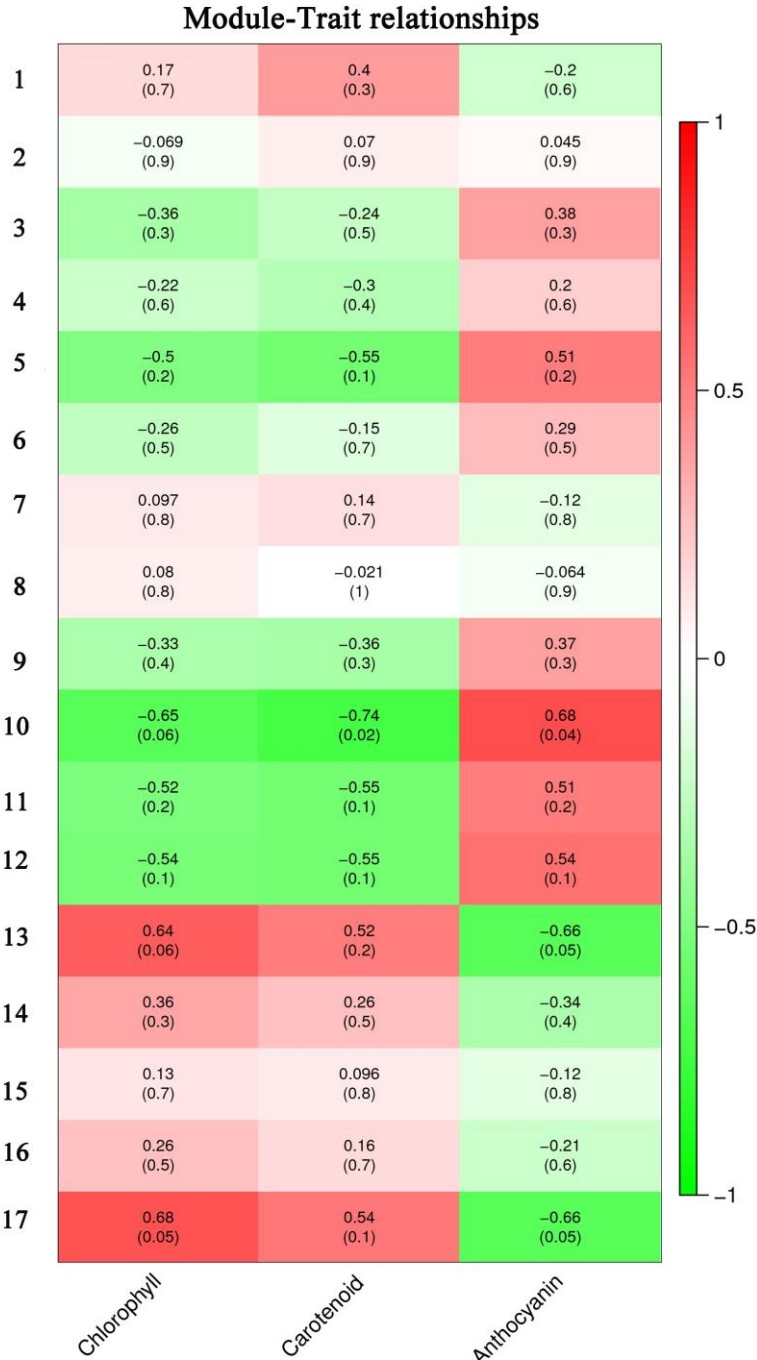

**Figure 3.** Pigment content and characteristic modules. Each column represents a physiological indicator, and each row represents a genetic module. The number in each grid represents the correlation between the module and the trait. The number in parentheses represents the *p*-value. The smaller the *p*-value, the stronger the significance of the representativeness and module correlation.

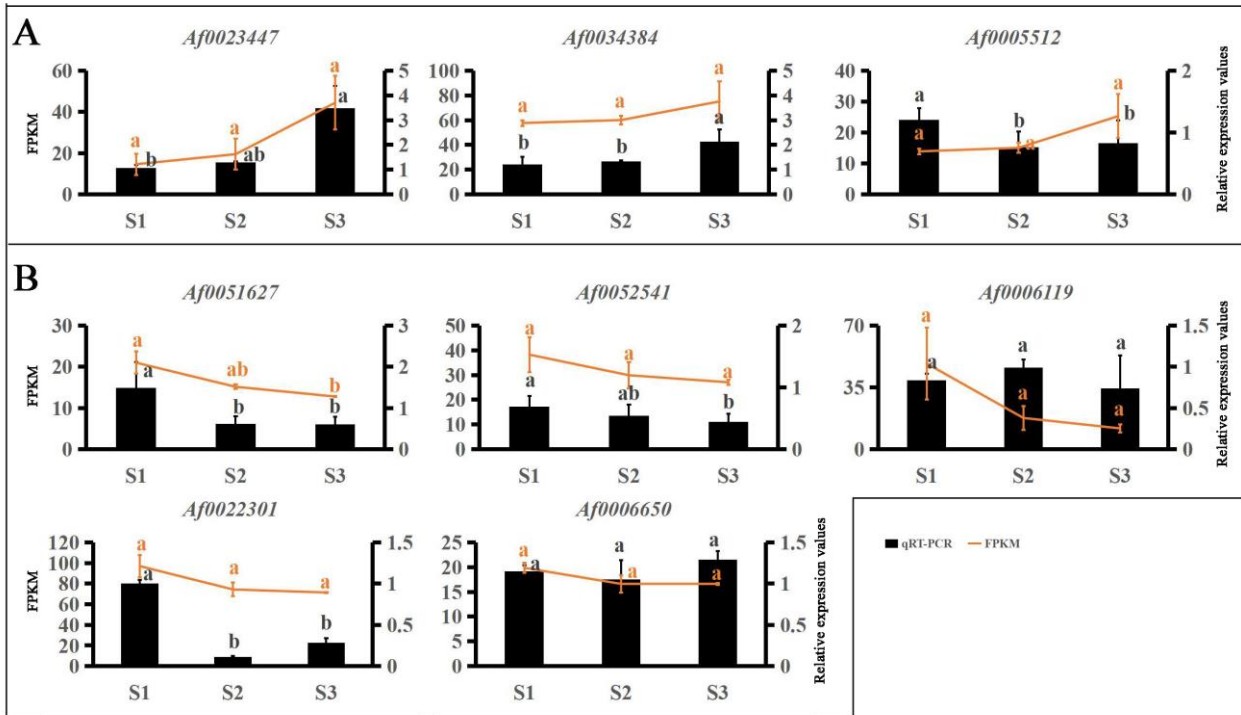

**Figure 4.** qRT-PCR validation of the transcriptome data results for hub genes. Expression levels of 8 genes and FPKM values. Different letters denote significant differences according to Tukey's test ($p < 0.05$). (**A**) Hub genes of carotenoid upregulation in the '17′ module. (**B**) Hub genes of anthocyanin in the '10′ module.

### 3.4. Co-Expression Networks Reveal the Regulatory Relationship between Carotenoids and Anthocyanins

Many TFs occur in the sequenced data of *A. fabri* leaves at different development stages. To further understand the regulatory network relationship between hub genes and TFs involved in carotenoid and anthocyanin production, TFs corresponding to two hub genes in the carotenoid upregulation module '17' were screened. In addition, TFs corresponding to two hub genes were also screened-out in the anthocyanin upregulation module '10'. Furthermore, hub genes and TFs were normalized and $R^2 > 0.6$ [11] was used to screen out combinations that could have an interaction relationship: *Af0051627* and *Af0033233*, *Af0052541* and *Af0049421*, *Af0034384* and *Af0010511* (Figures 5 and 6). Cytoscape was used to map the network regulation of hub genes and TFs in the two characteristic modules (Figure 7). One node in the carotenoid regulatory network connected four edges, and two nodes in the anthocyanin regulatory network connected four edges. The results of the gene network regulation map and qRT-PCR were used to develop a model based on the pigment content change and network regulation map during the color change of *A. fabri* leaves (Figure 8). During the transition from red to green leaves, TF *Af0010511* (NAC100) upregulated the expression of hub gene *Af0034384* (HSFB2A), leading to increased carotenoid content and gradually turning the leaves from red to green. However, during this transition, TFs *Af0033232* (NAC83) and *Af0049421* (WRKY24) downregulated the expression of the hub genes *Af0051627* (NMT1) and *Af0052541* (THY-1) respectively, leading to a decrease in anthocyanin content and the complete greening of the leaves. The candidate genes and TFs are described fully in Table S3.

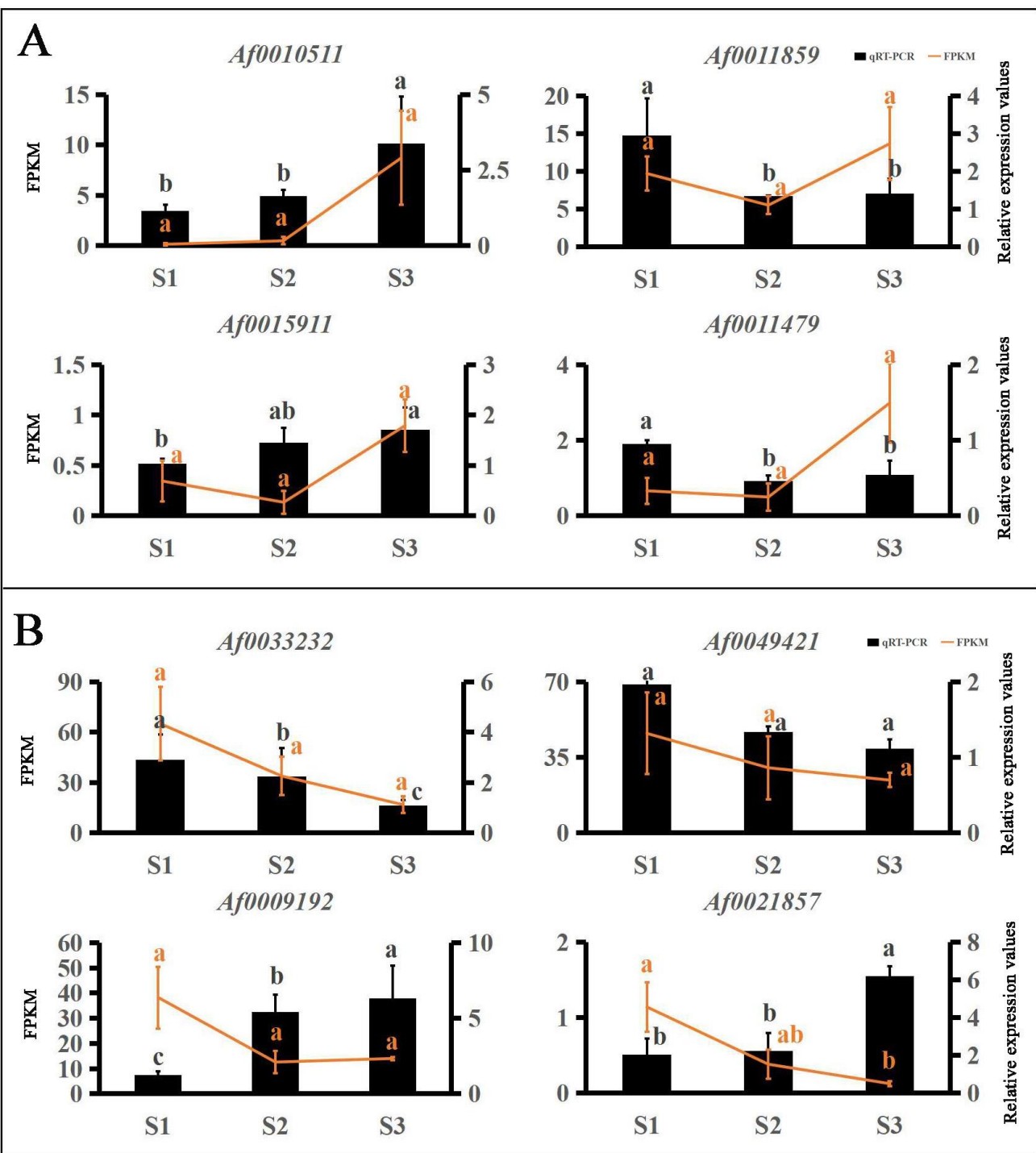

**Figure 5.** qRT-PCR validation of the transcriptome data results for TFs. Expression levels of 8 TFs and FPKM values. Different letters denote significant differences according to Tukey's test ($p < 0.05$). (**A**) TFs of carotenoid upregulation in the '17′ module. (**B**) TFs of anthocyanin in the '10′ module.

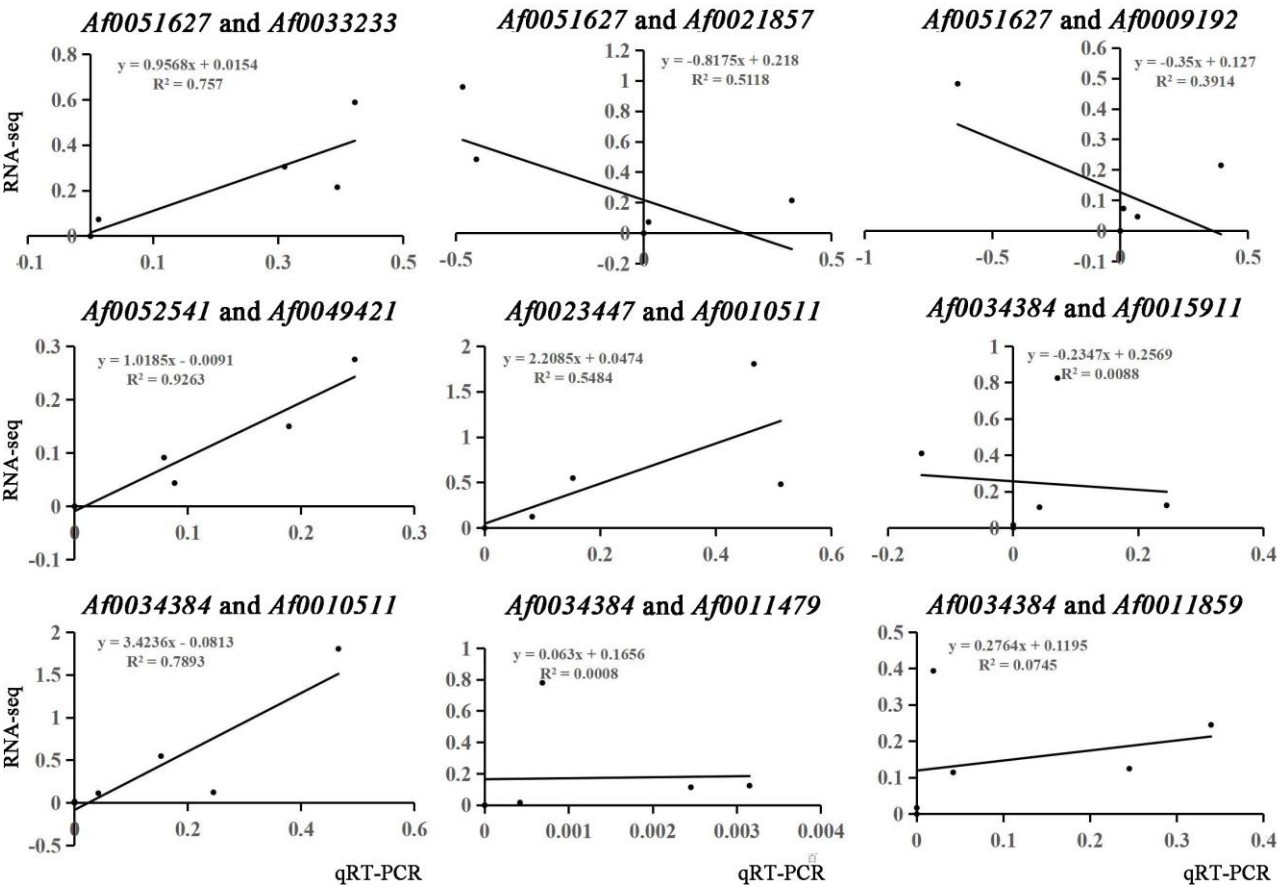

**Figure 6.** Correlation coefficient analysis of hub genes and edge genes in feature modules.

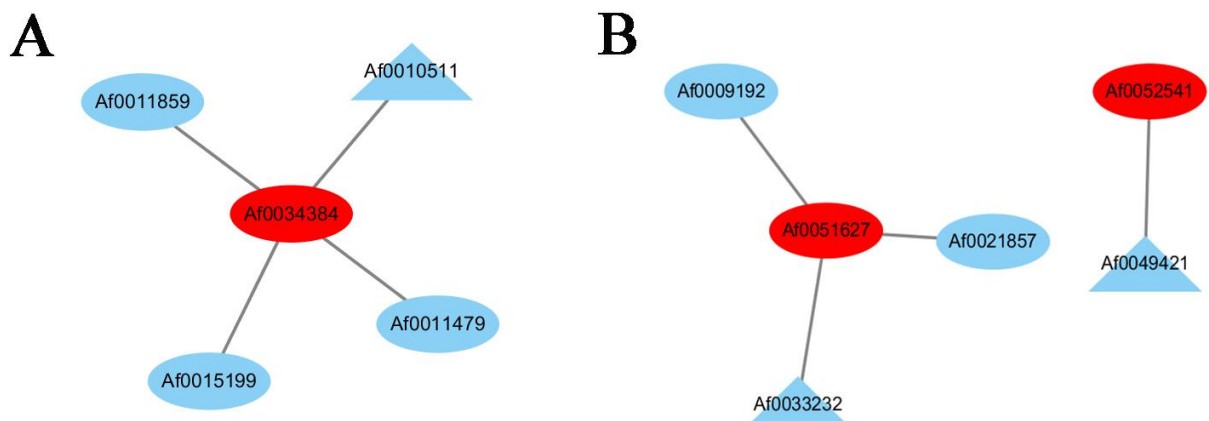

**Figure 7.** Regulatory networks of hub genes and edge genes in the feature modules. (**A**) Construction of a regulatory network of carotenoid u-regulation genes and transcription factors in the '17′ module. (**B**) Construction of a regulatory network of anthocyanin upregulation genes and transcription factors in the '10′ module. The hub genes are shown with red ellipses, the transcription factors are shown with blue ellipses, and the TFs with the highest correlation are shown with blue triangles.

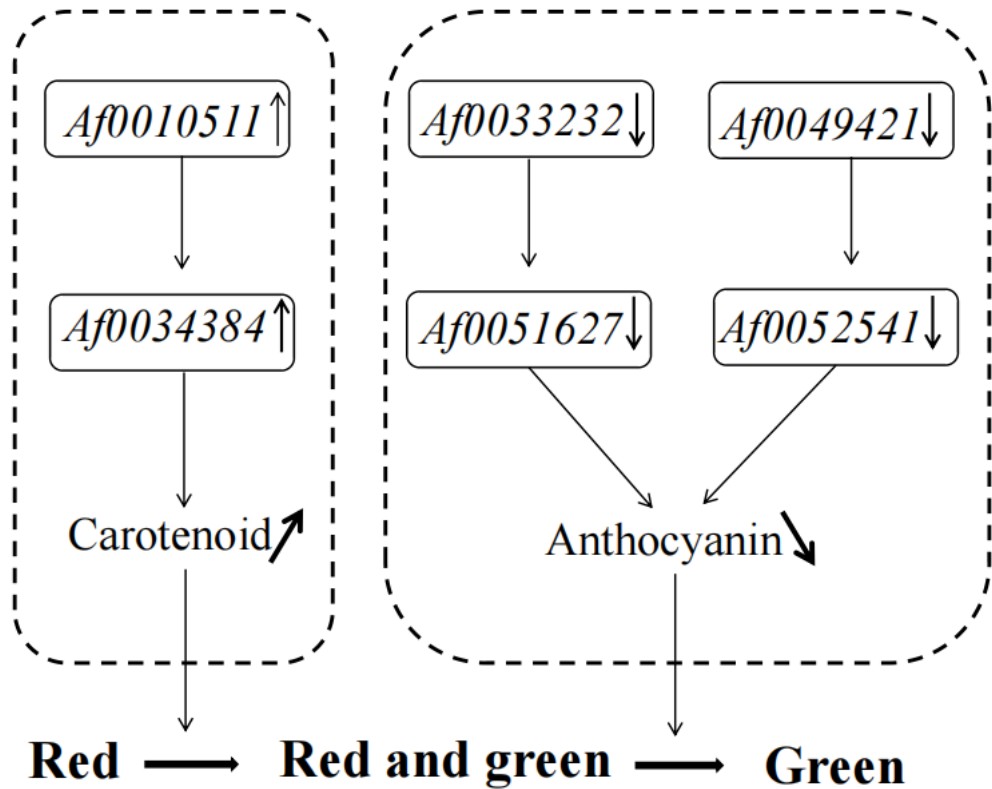

**Figure 8.** Proposed model of leaf color change in *A. fabri*.

## 4. Discussion

Leaf color is the most important ornamental character of *A. fabri*. The chlorophyll, carotenoid, and anthocyanin contents in leaves are important factors affecting leaf color change [33,34]. Young leaves of *A. fabri* gradually changed from red to red-green and then to green (Figure 1). An analysis of the anthocyanin content of red and green *Acer mono* leaves showed that the anthocyanin content of the red leaves was significantly increased compared with that of the green leaves [35]. A comparison of the leaves in the purple, purple-green, and green stages of *Paeonia lactiflora* showed that the chlorophyll content increased significantly and that the anthocyanin content decreased significantly from the purple to green stage [36]. Another study found that the anthocyanin content gradually decreased whereas the chlorophyll and carotenoid content increased in tree peony leaves, which transitioned from purplish red to half purple and half yellow-green to yellow-green [37]. We determined the pigment content of the leaves in the three color transition periods. As the leaves changed from red to red-green to green, the content of chlorophyll and carotenoid increased, whereas the anthocyanin content showed a gradual decline (Figure 2A). Thus, the changes in chlorophyll, carotenoid, and anthocyanin content led to a change in the leaf color of *A. fabri*, similar to the results of previous studies.

OPLS-DA integrates PLS-DA and orthogonal signal filtering technology, which can separate the information unrelated to pre-set values and classification from the original matrix to the greatest extent, to concentrate the most relevant factors into the first principal component. To further understand the pigments with a leading role in the changing of leaves from red to red-green and from red to green, three pigments were analyzed by OPLS-DA. The results of the highest values of OPLS-DA and VIP showed that carotenoids had a dominant role in the process of leaves from red to green, whereas anthocyanins were the main pigments in the process of leaves changing from red to green (Figure 2A and Table S2).

Given the lack of genomic information, there are few studies on the molecular mechanism of *A. fabri* leaf color change. The transcriptomes of some *Acer* spp. have been sequenced. For example, transcriptome sequencing was performed using young and mature *Acer truncatum* leaves, and many TFs involved in regulating anthocyanin biosynthesis

were obtained, providing a theoretical basis for the study of the molecular mechanism that determines the color change of maple leaves [38]. *Acer palmatum* with different color leaves (yellow and red) was sequenced, 18 structural genes were screened in the anthocyanin synthesis pathway, and 79 TFs involved in anthocyanin biosynthesis were identified [39]. Another study of red, yellow, and green *A. rubrum* leaves revealed 46 genes associated with anthocyanin synthesis and 69 genes associated with carotenoid synthesis [40]. In this study, transcriptome sequencing was performed on three phases of leaf color change in *A. fabri*. A total of 53,550 genes, including 838 TFs, were obtained through sequencing (Figure S1). These TFs might be involved in regulating the color change of *A. fabri* leaves, providing an avenue for future research.

WGCNA is a systematic biological method to describe the gene association patterns between different samples. It can identify gene modules with similar expression characteristics and conduct correlation analyses between the modules and biological traits to highlight core genes associated with the target traits [11,41,42]. The physiological and biochemical indexes and transcriptome data of different colored leaves of *O. fragrans* were analyzed and studied by WGCNA, and key genes regulating chlorophyll degradation and carotenoid metabolism pathways were screened out, providing new insights into the color formation mechanism of leaves in this species [12]. Similarly, in the current study, we performed WGCNA on three pigments and transcripts and identified two hub genes in the upregulated signature modules corresponding to carotenoids and anthocyanins: *Af00051627* and *Af0052541* (Figures 4 and 5, respectively). Studies have shown that TFs regulate the spatiotemporal expression and expression intensity of genes in metabolic pathways by binding to *cis*-acting elements of structural gene promoters, thus regulating the color of leaves. We screened TFs in the characteristic modules and identified the expression trend of hub genes and TFs by qRT-PCR, screening out three pairs of combinations that could have interaction relationships: *Af0051627* and *Af0033233*, *Af0052541* and *Af0049421*, and *Af0034384* and *Af0010511* (Figure 6). These results indicated that the hub genes were mainly regulated by the TFs and that they regulated the leaf color change in *A. fabri* by changing the carotenoid and anthocyanin content.

The metabolism of carotenoids and anthocyanins is important in plant leaf color change, and many TF family members directly or indirectly regulate carotenoids and anthocyanins, such as the bHLH TF family [43,44], WRKY TF family [45,46], and MYBs TF family [47,48], thus having important regulatory roles in leaf color formation. In the current study, three candidate TFs involved in regulating leaf color change, *Af0010511* (NAC), *Af0033232* (NAC), and *Af0049421* (WRKY), were identified and used for qRT-PCR analysis. WRKY has an important role in the transcriptional regulation of anthocyanin biosynthesis in plants [49]. Studies have reported that TF *WRKY75* could regulate anthocyanin accumulation in pears by activating promoters of DFR, UFGT, or MYB [50]. Similarly, in a study of *Actinidia chinensis*, the TF *WRKY44*, as a hub gene, was confirmed to regulate anthocyanin accumulation [51]. In the current study, TF *Af0049421* (WRKY) negatively regulated the expression of the hub gene *Af0052541* during the transition of red *A. fabri* leaves to red-green to completely green, resulting in a decrease in anthocyanin content (Figure 8), a result consistent with previous studies. Several studies have shown that most NAC TF family members, including TFs (*ORE1/NAC2*, *ANAC016*, *ANAC019*, *NAC29*, *ANAC046*, *ANAC055*, *ANAC072*, and *NAC092*), promoted chlorophyll degradation and leaf senescence [52–54]. Studies on transient overexpression in chrysanthemum leaves found that TF *CmNAC73* was a positive regulator of chlorophyll biosynthesis, which directly bound to the promoters of chlorophyll synthesis-related genes *HEMA1* and *CRD1* to regulate the color change of chrysanthemum leaves [14]. Differently from previous studies, and according to the hypothesis model, we concluded that, during leaf color change, two NAC TFs (*Af0010511* and *Af0033232*) regulated hub genes (*Af0034384* and *Af0051627*, respectively), which increased the carotenoid content and decreased the anthocyanin content (Figure 8). No NAC TFs were found to regulate the chlorophyll-related synthesis pathway, possibly because the gene regulation mechanism of leaf color formation in *A. fabri* differed from that in other

plants. We are now establishing the genetic transformation system of *A. fabri*, and the functions of the three TFs will be verified in future studies.

## 5. Conclusions

This is the first study of the transcriptome and physiology of *A. fabri* leaf color. The pigment content and transcriptome sequencing of maple leaf color at different developmental stages showed that the change from red to red-green was mainly regulated by carotenoids, whereas that from red–green to completely green was mainly regulated by anthocyanins. The correlation between the pigment and differential genes was analyzed by WGCNA. It was found that the hub genes *Af0034384* (HSFB2A), *Af0051627* (NMT1), and *Af0052541* (THY-1) were involved in the pigment regulatory network pathway and could be identified by the TFs *Af0010511* (NAC100), *Af0033232* (NAC83), and *Af0049421* (WRKY24); these genes changed the carotenoid and anthocyanin content of *A. fabri* leaves, regulating leaf color changes. These results have provided new insights for further studies of leaf color change in *A. fabri*.

**Supplementary Materials:** The following supporting information can be downloaded at: https://www.mdpi.com/article/10.3390/f14081638/s1. Figure S1: Gene number of each TF family; Table S1: Primer sequences used for qRT-PCR; Table S2: Principal component analysis of the three developmental stages of *A. fabri* leaf color. Table S3: The gene ID and descriptions of the selected genes.

**Author Contributions:** Conceptualization, L.W. and G.C.; methodology, G.L.; software, H.G.; validation, G.L., H.C. and Y.C.; formal analysis, G.L. and C.G.; investigation, H.C.; resources, G.L.; data curation, H.C.; writing—original draft preparation, G.L.; writing—review and editing, G.L.; visualization, H.G.; supervision, G.C.; project administration, G.L.; funding acquisition, G.L. All authors have read and agreed to the published version of the manuscript.

**Funding:** This research was funded by Color Leaf Tree Breeding and Cultivation of Provincial Long-term Scientific Research Base, grant number LYKJ[2020]26, Jiangsu Forestry Bureau, China; Study on Construction and Utilization of Germplasm Resource Bank of Colored Leaf Tree Species, grant number [2020KJ045], National Forestry and Grass Administration, China.

**Data Availability Statement:** No new data were created or analyzed in this study. Data sharing is not applicable to this article.

**Conflicts of Interest:** The authors declare that the research was conducted in the absence of any commercial or financial relationship that could be construed as potential conflict of interest.

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
