# Peer review of "Integrated Transcriptome and Biochemical Analysis Provides New Insights into the Leaf Color Change in Acer fabri"

_forests, doi:10.3390/f14081638_

Round 1

Reviewer 1 Report

This article describes how the gene expression of  Acer fabri leaves is associated with its color changes and how it is related to the carotenoid and anthocyanin genes.

Major

1. Did authors find which carotenoids are present in these leaves, are there any oxygenated carotenoids?

2. Also any carotenoid cleaving enzyme (CCDs) gene expression affected by the color change?

3. Line 157 materials and method section please use appropriate symbol for the temperature. 

Author Response

Comments and Suggestions for Authors

This article describes how the gene expression of Acer fabri leaves is associated with its color changes and how it is related to the carotenoid and anthocyanin genes.

Response: Thanks for your careful reading and fair evaluation of our work. We respect your comments and have dealt with each comment seriously in the following responses.

Major

  1. Did authors find which carotenoids are present in these leaves, are there any oxygenated carotenoids?

Response: Thanks for reminding us for this point. Total amount of carotenoids in these leaves was determined in this study, and no oxygenated carotenoids were found. This is really a great suggestion. We will conduct this work in our following study.

  1. Also any carotenoid cleaving enzyme (CCDs) gene expression affected by the color change?

Response: Thanks for reminding us for this point. Carotenoid cleaving enzyme (CCDs) gene expression can affect by the color change. However, we were not successful in screening for CCDs gene in our study

  1. Line 157 materials and method section please use appropriate symbol for the temperature. 

Response: I am very sorry for this. We have use appropriate symbol for the temperature. (line 161)

Reviewer 2 Report

This manuscript describes biochemical and gene expression changes associated with leaf colour change in Acer fabri. In general, the methods and initial analysis seems to be adequate for the experimental setup, and several similar studies in other species are referenced. However, in the manuscript, the differentially expressed genes are not described according to their annotation (except for the TFs in very general classifications). There is no comparison with the previous studies to identify if similar genes are differentially expressed, or no annotation of the genes identified in this study is presented in this manuscript. Addition of this information would improve the manuscript and provide a comparison with the existing scientific knowledge and literature.

Lines 67, 69 and elsewhere: The gene names e.g. CmNAC73 could also include a description of the gene annotation (e.g. transcription factor, biosynthetic, unknown, etc.).

The hub genes are only described in general terms, no gene annotations are given. Were these genes compared to previous studies (e.g. Chen et al. Transcriptome analysis based on a combination of sequencing platforms provides insights into leaf pigmentation in Acer rubrum. BMC Plant Biol. 2019, 19, 1-16.) to determine if they are orthologs or from the same gene family? This seems to be a simple task to compare the genes identified in this study with those identified in the Chen et al study and others, to detect any orthologous genes. 

In figure 3, why are the co-expression modules named after colours? Numbering them from 1-17 would be simpler and easier to find the relevant modules mentioned in the text.

In the ‘brown4’ module Af00051627 and Af0052541 were chosen as the hub genes, but why was Af0022301 rejected? Was is because of the smaller differential expression identified by RT-qPCR? These criteria could be included in the results or methods section

Figure 4 – Y axis should be labelled ‘Relative expression values’. There are no error bars for the RT-qPCR results. Were the RT-qPCR results tested for significance? The graphs are not sorted according to module – they should be placed according to module and possibly also labelled in the figure, this would enable a better understanding of the figure without referring back the text.

What is the basis of using a cut-off of R2 >0.6 to identify TF – gene expression interactions?

The meaning of Figure 7 is not clear. In the methods section and from Figure 6, the network analysis was done using the sequencing data. As mentioned above, the TF – gene expression interactions were identified by a cut-off of R2 >0.6. Why are the other TF-gene interactions included in Figure 7 – e.g. Af0051627 and Af0021857 and Af0051627 and Af0009192 (panel B), which have a negative correlation?

Minor comments

Line 32: Suggest to change “Plants the leaves of which change color with time…” to “Plants with leaves that change color over time…” or similar

Line 56:  Reference [3] is about anthocyanin accumulation in Acer mandshuricum

Lines 291-294: This is a repeat from the introduction and abstract.

Throughout the manuscript, refer to sequencing reads (not ‘readings’)

Table S1 is not mentioned in the text, Table S3 is mentioned before Table S2.

Line 32: Suggest to change “Plants the leaves of which change color with time…” to “Plants with leaves that change color over time…” or similar

Lines 291-294: This is a repeat from the introduction and abstract.

Throughout the manuscript, refer to sequencing reads (not ‘readings’)

Otherwise English language is of sufficient quality

Author Response

This manuscript describes biochemical and gene expression changes associated with leaf colour change in Acer fabri. In general, the methods and initial analysis seems to be adequate for the experimental setup, and several similar studies in other species are referenced. However, in the manuscript, the differentially expressed genes are not described according to their annotation (except for the TFs in very general classifications). There is no comparison with the previous studies to identify if similar genes are differentially expressed, or no annotation of the genes identified in this study is presented in this manuscript. Addition of this information would improve the manuscript and provide a comparison with the existing scientific knowledge and literature.

Response: Thanks for your careful reading and fair evaluation of our work. We respect your comments and have dealt with each comment seriously in the following responses.

  1. Lines 67, 69 and elsewhere: The gene names e.g. CmNAC73could also include a description of the gene annotation (e.g. transcription factor, biosynthetic, unknown, etc.).

Response: Thanks for reminding us for this point. We have added the description of those genes annotation. (line 69, 71, 73, 75, 76, 332, 334, 339 and 342)

  1. The hub genes are only described in general terms, no gene annotations are given. Were these genes compared to previous studies (e.g. Chen et al. Transcriptome analysis based on a combination of sequencing platforms provides insights into leaf pigmentation in Acer rubrum. BMC Plant Biol. 2019, 19, 1-16.) to determine if they are orthologs or from the same gene family? This seems to be a simple task to compare the genes identified in this study with those identified in the Chen et al study and others, to detect any orthologous genes. 

Response: Thanks for reminding us for this point. The descriptions of selected hub genes has shown in Table S3. We put hub genes sequence on the NCBI website for comparison. The results showed that two hub genes (Af0034384 (HSFB2A) and Af0051627 (NMT1)) have highest homology with the transcripome data published by Acer saccharum. Af0052541 (THY-1) had the highest homology with the genomic data published by Acer negundo.

Table S3. The gene ID and descriptions of selected genes.

Number

Gene ID

Descriptions

1

Af0034384

TXG70870.1 hypothetical protein EZV62_005805(HSFB2A)

2

Af0051627

TXG68515.1 hypothetical protein EZV62_003450(NMT1)

3

Af0052541

GAY66185.1 hypothetical protein CUMW_246740(THY-1)

4

Af0010511

TXG71357.1 hypothetical protein EZV62_006292(NAC100)

5

Af0033232

TXG54913.1 hypothetical protein EZV62_020169(NAC83)

6

Af0049421

TXG69628.1 hypothetical protein EZV62_004563(WRKY24)

  1. In figure 3, why are the co-expression modules named after colours? Numbering them from 1-17 would be simpler and easier to find the relevant modules mentioned in the text.

Response: Thanks for reminding us for this point. It is a good idea. We have numbered them from 1-17. The word of ‘brown4’ and ‘red’ has been replaced by ‘10’(line 210, 222, 231, 251, and 258) and ‘17’(line 204, 222, 229, 250, and 257) respectively throughout this manuscript.

  1. In the ‘brown4’ module Af00051627 and Af0052541 were chosen as the hub genes, but why was Af0022301 rejected? Was is because of the smaller differential expression identified by RT-qPCR? These criteria could be included in the results or methods section

Response: Thanks for reminding us for this point. We selected relative expression value of genes consistent with the expression trend of the transcriptome data as hub genes. So the gene Af0022301 was rejected. The sentences “According to the results of qRT-PCR, two final hub genes, Af00051627 and Af0052541, were identified (Figure 4).” have been changed to “Based on the correlation between expression trends of qRT-PCR and transcriptome data, two final hub genes Af00051627 and Af0052541 were identified,Af00051627 and Af0052541, were identified (Figure 4)” in line 210-212.

  1. Figure 4 – Y axis should be labelled ‘Relative expression values’. There are no error bars for the RT-qPCR results. Were the RT-qPCR results tested for significance? The graphs are not sorted according to module – they should be placed according to module and possibly also labelled in the figure, this would enable a better understanding of the figure without referring back the text.

Response: Thanks for your advice. The Figure 4-Y axis has been labelled ‘Relative expression values’. The error bars of RT-qPCR results has been added in Figure 4. At the same time, Figure 4 has been divided into two parts, which helps to understand the figure without referring back the text as well as the Figure 5.

Figure 4. qRT-PCR validation of the transcriptome data results for hub genes. Expression levels of 8 genes and FPKM values. Different letters denote significant differences according to Tukey’s test (P <0.05). (A) Hub genes of carotenoid u-regulation in the ‘17’ module. (B) Hub genes of anthocyanin in the ‘10’ module.

Figure 5 qRT-PCR validation of the transcriptome data results for TFs. Expression levels of 8 TFs and FPKM values. Different letters denote significant differences according to Tukey’s test (P <0.05). (A) TFs of carotenoid u-regulation in the ‘17’ module. (B) TFs of anthocyanin in the ‘10’ module.

  1. What is the basis of using a cut-off of R2 >0.6 to identify TF – gene expression interactions?

Response: Thanks for reminding us for this point. The method was adopted according to the previously published article(e.g. Gu et al. Integrated transcriptome and endogenous hormone analysis provides new insights into callus proliferation in Osmanthus fragrans. Sci Rep. 2022, 12, 1-13.)and it has been added on line 231.

  1. The meaning of Figure 7 is not clear. In the methods section and from Figure 6, the network analysis was done using the sequencing data. As mentioned above, the TF – gene expression interactions were identified by a cut-off of R2 >0.6. Why are the other TF-gene interactions included in Figure 7 – e.g. Af0051627 and Af0021857 and Af0051627 and Af0009192 (panel B), which have a negative correlation?

Response: Thanks for your advice.

We analyzed the correlation between qRT-PCR and FPKM (TFs and hub genes in the feature module) of Figure 4 and Figure 5. In general, the value of R2 is more closer to 1, the more likely there is a regulatory relationship between them. Our results showed that Af0051627 and Af0021857, Af0051627 and Af0009192 had low correlation coefficients and were therefore not listed in the range of candidate genes.

Minor comments

  1. Line 32: Suggest to change “Plants the leaves of which change color with time…” to “Plants with leaves that change color over time…” or similar

Response: Thanks for your advice. The sentence “Plants the leaves of which change color with time Plants the leaves of which change color with time” has been replaced by “Plants with leaves that change color over time exist widely in nature and have unique ornamental value [1,2].”(line 33)

  1. Line 56:  Reference [3] is about anthocyanin accumulation in Acer mandshuricum

Response: I am sorry for it. The reference [3] has been changed. (line 389)

Refences:

  1. Grotewold, E. The genetics and biochemistry of floral pigments. Annu RevPlant Biol. 2006, 57, 761-780.

  1. Lines 291-294: This is a repeat from the introduction and abstract.

Response: I am very sorry for this mistake. The sentence of “Acer palmatum cultivars with yellow and red leaves were selected for high-throughput sequencing. This identified 18 structural genes involved in anthocyanin biosynthesis and were related to anthocyanin accumulation; 46 MYBs and 33 bHLHs were identified as being involved in regulating anthocyanin biosynthesis [40].” has been replaced by “Acer palmatum with different color leaves (yellow and red) was sequenced, 18 structural genes were screened in the anthocyanin synthesis pathway, and 79 TFs involved in anthocyanin biosynthesis were identified [40].”. (line 296-299)

  1. Throughout the manuscript, refer to sequencing reads (not ‘readings’)

Response: I am sorry for it. The word of ‘readings’ has been replaced by ‘sequencing reads’ in the whole manuscript. (line 134, 190 and 191)

  1. Table S1 is not mentioned in the text, Table S3 is mentioned before Table S2.

Response: Thanks for reminding us for this point. We examined our manuscript carefully. Table S1 is mentioned in the text (line 162). Table S3 (line 246) is mentioned after Table S2 (line 182 and 290).

  1. Otherwise English language is of sufficient quality

Response: Thanks for your advice. We have retouched the language according to your advice. That language has been edited by MDPI.
